# Comparison of the Pathogenicity of Classical Swine Fever Virus Subgenotype 2.1c and 2.1d Strains from China

**DOI:** 10.3390/pathogens9100821

**Published:** 2020-10-07

**Authors:** Genxi Hao, Huawei Zhang, Huanchun Chen, Ping Qian, Xiangmin Li

**Affiliations:** 1State Key Laboratory of Agricultural Microbiology, Huazhong Agricultural University, Wuhan 430070, China; chzhx5210@webmail.hzau.edu.cn (G.H.); ZHW@mail.hzau.edu.cn (H.Z.); chenhch@mail.hzau.edu.cn (H.C.); 2Laboratory of Animal Virology, College of Veterinary Medicine, Huazhong Agricultural University, Wuhan 430070, China; 3Key Laboratory of Development of Veterinary Diagnostic Products, Ministry of Agriculture, Wuhan 430070, China; 4Key Laboratory of Preventive Veterinary Medicine in Hubei Province, The Cooperative Innovation Center for Sustainable Pig Production, Wuhan 430070, China

**Keywords:** classical swine fever virus (CSFV), subgenotype 2.1c, subgenotype 2.1d, pathogenicity, China

## Abstract

Classical swine fever (CSF) caused by classical swine fever virus (CSFV) is a highly contagious and devastating disease. The traditional live attenuated C-strain vaccine is widely used to control disease outbreaks in China. Since 2000, subgenotype 2.1 has become dominant in China. Here, we isolated subgenotype 2.1c and 2.1d strains from CSF-suspected pigs. The genetic variations and pathogenesis of subgenotype 2.1c and 2.1d strains were investigated experimentally. We aimed to evaluate and compare the replication characteristics and clinical signs of subgenotype 2.1c and 2.1d strains with those of the typical highly virulent CSFV SM strain. In PK-15 cells, the three CSFV isolates exhibited similar replication levels but significantly lower replication levels compared with the CSFV SM strain. The experimental animal infection model showed that the pathogenicity of subgenotype 2.1c and 2.1d strains was less than that of the CSFV SM strain. According to the clinical scoring system, subgenotype 2.1c (GDGZ-2019) and 2.1d (HBXY-2019 and GXGG-2019) strains were moderately virulent. This study showed that the pathogenicity of CSFV field strains will aid in the understanding of CSFV biological characteristics and the related epidemiology.

## 1. Introduction

Classical swine fever (CSF) is a highly contagious disease of pigs caused by classical swine fever virus (CSFV) [1]. CSF also causes great harm to the pig industry. Pigs are the natural hosts of CSFV and pigs of various breeds or ages can be infected. CSFV is a positive sense single-stranded RNA virus and a member of the genus Pestivirus within the family Flaviviridae [2]. The CSFV genome contains a large open reading frame that encodes four structural proteins (C, Erns, E1 and E2) and eight nonstructural proteins (Npro, p7, NS2, NS3, NS4A, NS4B, NS5A and NS5B) [3].

The E2 protein is the main structural protein of CSFV and is highly variable among isolates; it induces the neutralizing antibodies and shows a relationship with virulence [4,5]. On the basis of the E2 gene, CSFV isolates can be divided into three genotypes (1, 2 and 3) and are further subdivided into 11 subgenotypes (1.1–1.4, 2.1–2.3 and 3.1–3.4) [6]. CSFV outbreaks caused by genotype 2 have been increasing in Europe and Asia [2,3,7,8]. Given this situation, the genetic evolution of CSFV has been analyzed in detail and subgenotype 2.1 isolates have been further classified into 10 clades (2.1a–2.1j). Phylogenetic analysis indicates that CSFV in pigs in China includes subgenotypes 1.1, 2.1, 2.2 and 2.3b [6,9]. Since 2000, subgenotype 2.1 has become dominant in China. Among all subgenotypes of 2.1, subgenotypes 2.1c and 2.1d are currently the most widely prevalent in China [1,6,10]. The field virulence of CSFV is inconsistent with its genotype [11,12]. No consensus has been reached with regard to the virulence of pandemic CSFV strains.

In this study, the characterization of the CSFV isolates from three farms in Hubei, Guangxi and Guangdong provinces were evaluated. To investigate the virulence of subgenotype 2.1c and 2.1d isolates, we compared the pathogenicity of subgenotype 2.1 strains and the subgenotype 1.1 SM strain that is the high-virulence strain in China.

## 2. Results

### 2.1. Virus Isolation

RT-PCR assays were performed with the amplified E2 gene fragments of CSFV to detect clinical samples. Positive clinical CSFV samples were inoculated into PK-15 cells. The inoculated cells were passaged successively for the fifth generation. RT-PCR assay and indirect IFA showed positive results (Figure 1). CSFV strains were isolated from positive samples. The three CSFV isolates, which were obtained from Hubei, Guangdong and Guangxi, were named as HBXY-2019, GDGZ-2019 and GXGG-2019, respectively.

### 2.2. Phylogenetic Analysis of the E2 Gene

The nucleotide sequences of the three isolated CSFV strains were compared with those of the other CSFV strains in GenBank. Sequence analysis revealed that the E2 gene of the three isolated strains showed 81.1–98.4% nucleotide similarity with other Chinese strains. The E2 AA sequence homologies among three isolated strains ranged from 95.6% to 99.5% and all isolates shared high similarities of 96.4–99.7% with CSFV from subgenotype 2.1 and lower identities with genotype 1 (89.3–90.4%).

The phylogenetic trees of the full-length E2 gene were constructed by using MEGA7.0 (https://www.megasoftware.net/). Phylogenetic analysis indicated that the three isolated CSFV strains belonged to subgenotype 2.1. CSFV HBXY-2019 and GXGG-2019 isolates were grouped into subgenotype 2.1d and GDGZ-2019 was grouped into subgenotype 2.1c (Figure 2).

### 2.3. Comparison of the AA Mutations of the E2 Gene in Three CSFV Isolated Strains

As shown in Figure 3, 44 mutated AAs were observed between the three CSFV isolated strains and subgenotype 1.1, and 15 mutated AAs were detected among GDGZ-2019, HBXY-2019 and GXGG-2019. Two AAs were mutated in HBXY-2019 and GXGG-2019 (K197M and R303K). The E2 proteins of different genotypes contained high levels of mutated AAs, indicating that E2 proteins were mutant viral proteins (Figure 3 and Table 1).

### 2.4. Virus Proliferation of CSFV Isolated Strains

We performed one-step growth experiments to analyze the replication characteristics of the three isolated CSFV strains. As shown in Figure 4A, no significant difference was observed among the three isolated CSFV strains. However, the CSFV SM strain exhibited a significantly higher replication level than the three isolated CSFV strains. At 72 hpi, the average CSFV SM viral titer was 10^7.5^ TCID_50_/mL, whereas GDGZ-2019, HBXY-2019 and GXGG-2019 viral titers were 10^6.3^, 10^6.7^ and 10^6.1^ TCID_50_/mL, respectively (Figure 4).

### 2.5. Pathogenicity Analysis of CSFV Isolated Strains

All pigs in the CSFV SM group showed typical high fever post-challenge. The average rectal temperature of pigs infected with CSFV SM exceeded 41.0 °C at 3 dpi and a high body temperature was continually observed until death. All the pigs in the CSFV SM group died at 7–13 dpi and displayed typical CSF symptoms. The average rectal temperature of pigs in the GDGZ-2019, HBXY-2019 and GXGG-2019 groups rose and exceeded 40.0 °C at 3 or 4 dpi (Figure 5A). The temperature of two pigs in the HBXY-2019 and GXGG-2019 groups exceeded 41 °C at 7 or 11 dpi. Three pigs in the GDGZ-2019 group had a fever temperature of more than 41 °C at 5, 8 and 11 dpi. The temperatures of all pigs normalized at 18 dpi. Two pigs in the HBXY-2019 and GXGG-2019 group died at the end of the experiment, whereas three dead pigs were noted in the GDGZ-2019 group (Figure 5B).

All pigs in the CSFV SM, GDGZ-2019, HBXY-2019 and GXGG-2019 groups displayed clinical signs of CSFV infection (Figure 5C). Clinical signs were the most severe in the CSFV SM group but developed slowly and were less severe in the GDGZ-2019, HBXY-2019 and GXGG-2019 groups. However, infected pigs with rectal temperatures over 41 °C displayed more clinical symptoms than those in the GDGZ-2019, HBXY-2019 and GXGG-2019 groups.

According to the CS system, the clinical score of the CSFV SM group was higher than that of the other group. The peak CS value of the pigs infected with CSFV SM was 26 and the CS value of each pig exceeded 15. The CS values of all pigs in the GDGZ-2019, HBXY-2019 and GXGG-2019 groups were all less than 15 (Figure 5C). Three out of five pigs in the GDGZ-2019 group had scores higher than 10 at 12 dpi and two out of five pigs in the HBXY-2019 and GXGG-2019 groups also had scores higher than 10. On the basis of mean clinical scores, the GDGZ-2019, HBXY-2019 and GXGG-2019 strains were defined as moderately virulent compared with the highly virulent CSFV SM strain.

Viremia was detected through virus isolation techniques. Viremia in blood was detected in all pigs of the CSFV SM, GDGZ-2019, HBXY-2019 and GXGG-2019 groups (Figure 5D). The viral titer of pigs infected with CSFV SM gradually increased throughout the experiment. The viral titer of pigs infected with the GDGZ-2019, HBXY-2019 and GXGG-2019 strains peaked at 15 dpi. Then, viremia rapidly declined in the GDGZ-2019, HBXY-2019 and GXGG-2019 groups.

### 2.6. Hematological Data

CSFV infection may induce leukopenia and immunosuppression. Blood samples were collected at different times post-challenge for the hematology test. The leukocyte and platelet counts were low in all CSFV-infected pigs (Figure 6). Leukopenia and thrombocytopenia were significantly severe at 6 and 12 dpi (*p* < 0.05) in CSFV SM-infected pigs. CSFV SM-infected pigs showed reduced leukocyte and PLT counts until death. However, the leukopenia and thrombocytopenia of GDGZ-2019, HBXY-2019 and GXGG-2019-infected pigs were significantly less than those in CSFV SM-infected pigs. Differences among the GDGZ-2019, HBXY-2019 and GXGG-2019 groups were not significant. Leukocyte counts and platelet counts in GDGZ-2019, HBXY-2019 and GXGG-2019-infected pigs were reduced to a minimum at 12 dpi. Then, the leukocyte counts and platelet counts of the pigs infected with the three strains gradually recovered and returned to normal at the end of the experiment.

## 3. Discussion

The traditional live attenuated C-strain vaccine is widely used in the world and plays a critical role in controlling CSF in multiple countries [13]. The immune effect of the C-strain is safe and effective. CSF has been controlled and eradicated in many countries. Vaccination and differential diagnosis are effective ways to eradicating CSF. However, the C-strain vaccine cannot serologically discriminate between vaccinated animals and infected animals [5,14]. Therefore, the classical attenuated vaccine encounters challenges in eradicating the CSF epidemic. At present, the C-strain is widely used to control disease outbreaks in China and frequent vaccinations are performed in pig farms. For the moment, the numbers of immunization failure and atypical clinical signs of CSF have been observed in clinical practice [1,6,9]. No mass outbreak of CSF has been recorded in recent years, whereas sporadic cases have been reported in C-strain-vaccinated farms in many regions of China.

In this study, three CSFV isolates were isolated from CSF-suspected pigs. The E2 sequences of isolated strains were compared with other CSFV strains in GenBank. One isolate was clustered into subgenotype 2.1c and the other two isolates were clustered into subgenotype 2.1d (Figure 2). These isolates shared high similarities of 96.4–99.7% with CSFV from subgenotype 2.1 and low identities with genotype 1 (89.3–90.4%). The E2 protein of these isolates had 44 mutated AAs compared with that of subgenotype 1.1. The WH303 epitope (TAVSPTTLR) of the E2 protein is an immunodominant epitope among CSFV strains. This epitope also exists in the three CSFV isolates, showing no genetic variability.

We compared the replication capability of subgenotypes 2.1c and 2.1d via one-step growth experiments and evaluated their pathogenicity in weaned piglets. Three CSFV isolates exhibited similar replication levels in PK-15 cells. However, these isolates exhibited significantly lower replication levels than the CSFV SM strain (*p* < 0.05). The results of animal experiments indicated that subgenotypes 2.1c (GDGZ-2019) and 2.1d (HBXY-2019 and GXGG-2019) were less pathogenic than the CSFV SM strain. No significant difference was noted among the three CSFV isolates. The virulence of subgenotypes 2.1c (GDGZ-2019) and 2.1d (HBXY-2019 and GXGG-2019) was assessed on the basis of the CS system. Subgenotypes 2.1c (GDGZ-2019) and 2.1d (HBXY-2019 and GXGG-2019) were moderately virulent (5 < CS ≤ 15) compared with the highly virulent SM strain, a known reference strain that belongs to subgenotype 1.1. Several studies have reported that genotype 2 of CSFV is less virulent than genotype 1. We also observed that subgenotypes 2.1c (GDGZ-2019) and 2.1d (HBXY-2019 and GXGG-2019) had moderate pathogenicity; this observation is consistent with the findings of other studies [1,10,15,16]. The moderate clinical symptoms caused by the CSFV isolates observed in the present study were consistent with the atypical clinical signs of CSF observed in clinical practice. Previous studies have demonstrated that the E2 protein is a determinant of virulence [4,17,18]. The substitution or mutation of E2 leads to viral attenuation [18]. A recent study showed that AA T56I and M290K substitutions, especially the M290K mutation, in E2 protein increase virus pathogenicity [19]. A similar mutation was detected among three CSFV isolates (Figure 3). However, CSFV virulence is also affected by other gene mutations [20,21].

At present, subgenotype 2.1 strains are the dominant pandemic strains in China [6,10,22]. Three CSFV isolates were isolated from different provinces of China in the present study. These CSFV isolates belong to two different subgenotypes (2.1c and 2.1d). We compared the pathogenicity of subgenotype 2.1c and 2.1d strains via the intranasal route in weaned piglets. After pigs were infected with CSFV 2.1 isolates, there were some marked characteristic changes, such as hemocytopenia, especially leukopenia and thrombocytopenia. CSFV SM-infected pigs showed reduced leukocyte and PLT counts until death. However, leukopenia and thrombocytopenia of GDGZ-2019, HBXY-2019 and GXGG-2019-infected pigs were significantly less than those in CSFV SM-infected pigs and gradually recovered and went back to normal at the end of the experiment. CSFV 2.1c (GDGZ-2019) and 2.1d (HBXY-2019 and GXGG-2019) strains are moderate virulent strains, similar to the published results on subgenotype 2.1 strains. This study shows the necessity of monitoring the molecular epidemiology and the etiological characteristics of the epidemic CSFV 2.1 isolates, which may help us to understand the CSFV 2.1 isolates’ biological characteristics and control the CSF outbreaks.

## 4. Materials and Methods

### 4.1. Sample Collection

Ten clinical samples (tonsils, lymph nodes and spleen) were collected from three pig farms in China (Hubei, Guangdong and Guangxi) in 2019. The RNA of clinical samples was isolated by using TRIzol reagent (Invitrogen, Carlsbad, California, USA) in accordance with the manufacturer’s instructions. RNA was then quantified and reverse-transcribed by using a First Strand cDNA Synthesis Kit (TOYOBO, Osaka City, Osaka Prefecture, Japan) in accordance with the manufacturer’s instructions. All samples were amplified by reverse transcription-polymerase chain reaction (RT-PCR) by using previously described specific primers, which were used for the CSFV E2 full-sequence gene [5]. The amplified gene segments were then ligated into a pEASY^®^-blunt cloning vector (Transgen Biotech, Beijing, China). Positive clones were sequenced by using Sanger Sequencing Technology and submitted to NCBI (GenBank accession number: MT422346, MT422347, MT422348).

### 4.2. Cells and Antibodies

Porcine kidney cells (PK-15; ATCC, CCL-33) were grown in Dulbecco’s modified essential medium (DMEM; Invitrogen, Waltham, MA, USA) containing 10% fetal bovine serum (FBS) (Gibco, Waltham, MA, USA) at 37 °C in a humidified 5% CO_2_ incubator. E2-specific monoclonal antibodies were prepared in our laboratory. Alexa Fluor 488 goat anti-mouse secondary antibody was obtained from Life Technologies, USA.

### 4.3. CSFV Isolation

Virus isolation was conducted in PK-15 cells maintained in DMEM (Invitrogen, Waltham, MA, USA) supplemented with 10% FBS (Invitrogen) at 37 °C. Clinical samples were ground into suspensions and clarified through centrifugation. The supernatant was passed through a sterile 0.22 µm filter and inoculated into PK-15 cells. After 60 h of incubation, the supernatant was harvested. RT-PCR with CSFV E2-specific primers verified that the viral isolate was CSFV.

### 4.4. Indirect Immunofluorescence Assay

PK-15 cells were seeded into 24-well plates and separately infected with the CSFV strains. At 36 h post-infection, the cells were fixed in 4% paraformaldehyde for 20 min, permeabilized with 0.1% Triton X-100 at room temperature for 10 min and blocked with 2% bovine serum albumin. The cells were subsequently incubated with AH09 E2-specific monoclonal antibodies (1:200 dilution in our laboratory) and the fixed cells were incubated with Alexa Fluor 488 goat anti-Mouse secondary antibodies (1:1000 dilution, Invitrogen). Fluorescence was observed under an Olympus IX73 fluorescent microscope.

### 4.5. Phylogenetic Analysis

The amino acid (AA) sequences of the three isolates were analyzed by using the Clustal W method of Lasergene (Version 7.1) (DNASTAR Inc., Madison, WI, USA). The phylogenetic trees of full-length E2 genes were constructed through the neighbor-joining method and the maximum composite likelihood model was established by using 1000 replicates with bootstrap values.

### 4.6. One-Step Growth Curve

The confluent monolayers of PK-15 cells in T25 flasks were inoculated with the CSFV SM, HBXY-2019, GDGZ-2019, or GXGG-2019 strains at a multiplicity of infection equal to 0.1. After incubation for 1 h at 37 °C, the inoculated cells were washed twice with phosphate-buffered saline. Then, a fresh medium containing 2% FBS was added. The infected cells were further cultured in an incubator at 37 °C. The culture supernatant was harvested by centrifugation at 0, 12, 24, 36, 48, 60 and 72 h. The viral titers were calculated as median tissue culture infective dose (TCID_50_) and determined by IFA, which was performed as described above [1].

### 4.7. Animal Experiment

The animal experiment was approved by the Research Ethics Committee of College of Veterinary Medicine, Huazhong Agricultural University, Hubei, China (No.20190526). Twenty 8-week-old three-breed cross pigs were purchased from the experimental farm of Huazhong Agricultural University and randomly divided into four groups. All pigs were confirmed to be seronegative for CSFV by neutralization test and RT-PCR. The four groups of pigs were housed in the negative-pressure facility of Wuhan Keqian Biology Co., Ltd (Wuhan, China) and were placed in separate rooms to avoid cross infection.

Pigs in group A (control group) were inoculated intranasally with 2 × 10^6^ TCID_50_ of virulent CSFV SM strain. Pigs in group B were inoculated intranasally with 2 × 10^6^ TCID_50_ of HBXY-2019 strain. Pigs in group C were inoculated intranasally with 2 × 10^6^ TCID_50_ of GDGZ-2019 strain. Pigs in group D were inoculated intranasally with 2 × 10^6^ TCID_50_ of GXGG-2019 strain. Following challenge, the rectal temperature and clinical signs (liveliness, body tension, body shape, breathing, neurological signs, conjunctivitis, appetite, defecation and so on) were monitored daily. A rectal temperature above 40.0 ℃ is considered an indication of fever. The clinical scoring (CS) system was used to evaluate the virulence of CSFV in accordance with previously established standards [23]. Clinical scores were evaluated based on 10 clinical parameters [23]. Each parameter was calculated as follows: normal, 0 points; slightly altered, 1 point; distinct clinical signs, 2 points; severe CSF symptom, 3 points. The maximum total score per pig was 30.

### 4.8. Routine Blood Tests

Anticoagulated blood was collected every 3 days post-challenge until the trial ended. Leukocyte counts and platelet counts were determined using a Mindray BC-2800 Vet analyzer (Shenzhen Mindray Bio-Medical Electronics Co., Shenzhen, China).

### 4.9. Statistical Analysis

Statistical analyses were performed using one-way ANOVA in GraphPad Prism software (GraphPad Software Inc., La Jolla, CA, USA). *p* < 0.05 was considered statistically significant.

## Figures and Tables

**Figure 1 pathogens-09-00821-f001:**
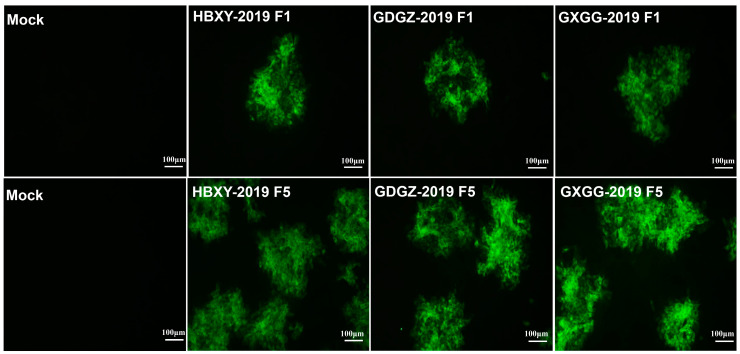
Identification of isolated CSFV strains. IFA of PK-15 cells infected with isolated CSFV strains at 36 h post-infection.

**Figure 2 pathogens-09-00821-f002:**
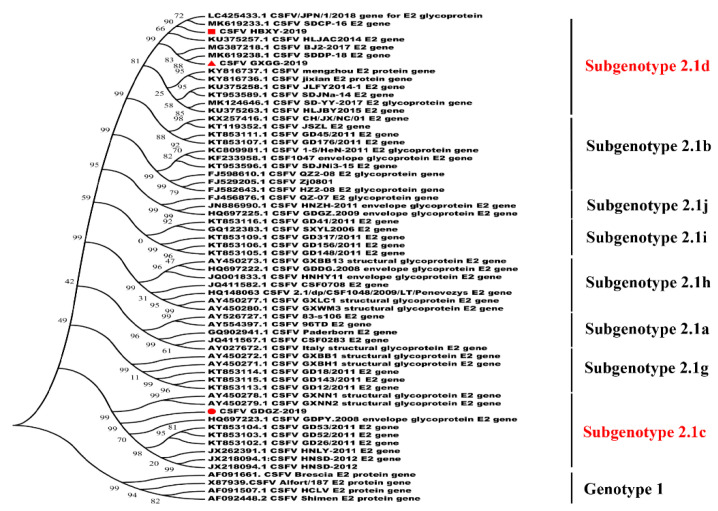
Phylogenetic analysis based on the full-length E2 gene of the isolated virus. Phylogenetic trees were constructed using MEGA 7.0.18 software with the neighbor-joining method (1000 bootstrap replicates). The isolated virus is marked in red.

**Figure 3 pathogens-09-00821-f003:**
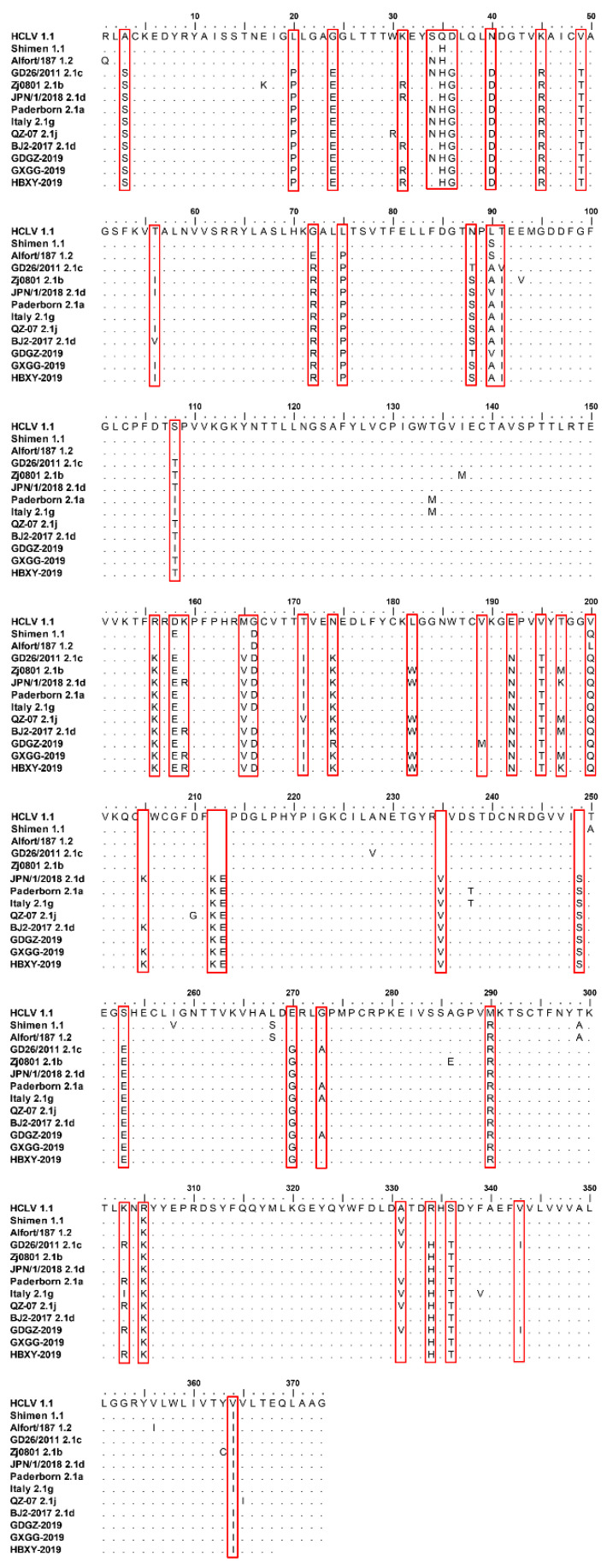
Amino acid sequence analysis of E2 genes of the three CSFV isolates. The special sites of AA mutation of these isolates are marked as red boxes.

**Figure 4 pathogens-09-00821-f004:**
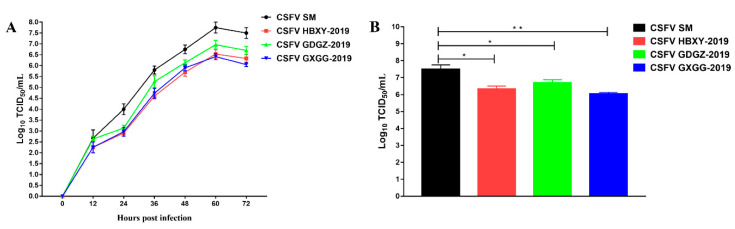
Comparison of the characteristics of the three CSFV isolates in PK15 cells. (**A**) One-step growth curves of the three CSFV isolates. (**B**) At 72 h post infection, the virus titers were determined in PK15 cells. Data represent the mean ± SEM from three independent experiments. * *p* < 0.05; ** *p* < 0.01.

**Figure 5 pathogens-09-00821-f005:**
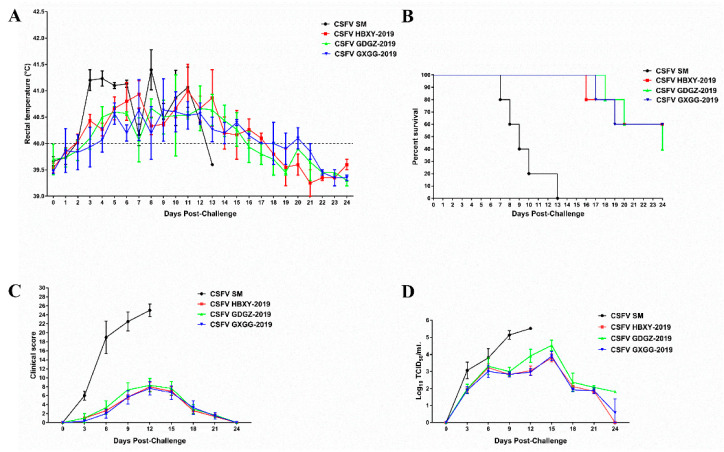
Rectal temperatures (**A**), survival rates (**B**), clinical scores (**C**) and viremia levels (**D**) of the pigs infected with the three CSFV isolates. Virus titers in blood are expressed as the mean log_10_TCID_50_/mL. All data are expressed as mean ± SEM.

**Figure 6 pathogens-09-00821-f006:**
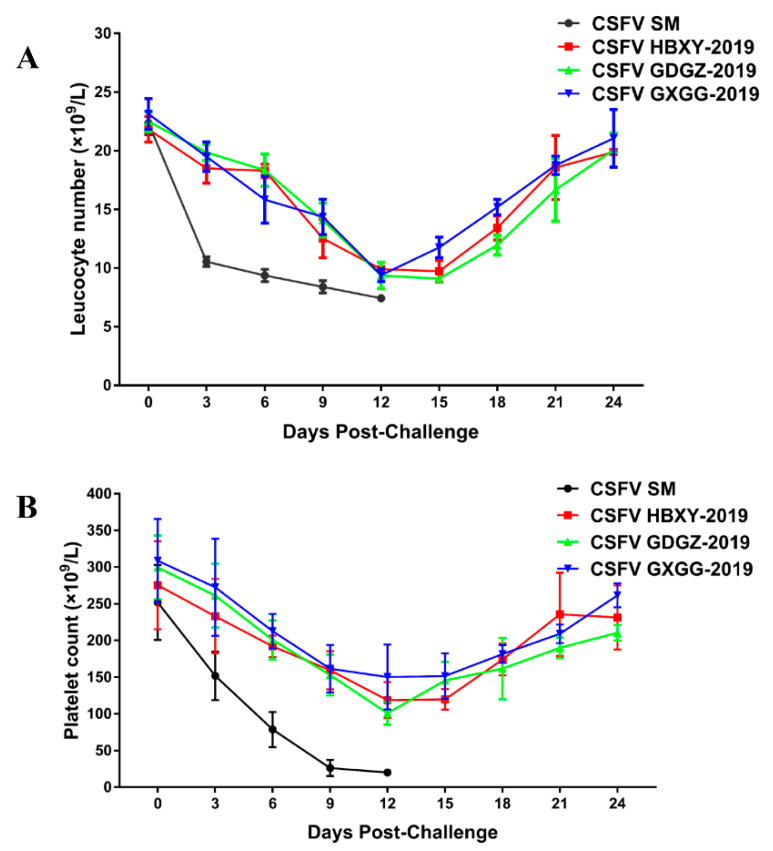
Hematological data of clinical samples from animals infected with the three new CSFV isolates. The leukocyte numbers (**A**) and platelet counts (**B**) are expressed as numbers/L of blood. Data are presented as the mean ± SEM.

**Table 1 pathogens-09-00821-t001:** The difference in the amino acid of E2 gene of the isolated virus.

Positions (AA)	HCLV	GDGZ-2019	GXGG-2019	HBXY-2019
3	A	S	S	S
20	L	P	P	P
24	G	E	E	E
31	K	K	R	R
34	S	N	S	S
35	Q	H	H	H
36	D	G	G	G
40	N	D	D	D
45	K	R	R	R
49	V	T	T	T
56	T	T	I	I
72	G	R	R	R
75	L	P	P	P
88	N	T	S	S
90	L	V	A	A
91	T	I	I	I
108	S	I	T	T
156	R	K	K	K
158	D	E	E	E
159	K	K	R	R
165	M	V	V	V
166	G	D	D	D
171	T	I	I	I
174	N	R	K	K
182	L	L	W	W
192	E	N	N	N
195	V	T	T	T
197	T	T	M	K
200	V	Q	Q	Q
205	R	R	K	K
212	D	N	N	N
213	G	E	E	E
235	I	V	V	V
249	R	S	S	S
253	S	E	E	E
270	E	G	G	G
273	G	A	G	G
290	M	R	R	R
303	K	R	K	R
305	R	K	K	K
331	A	V	A	A
334	R	H	H	H
336	S	T	T	T
343	V	I	V	V

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
