# Peer review of "Comparison of the Pathogenicity of Classical Swine Fever Virus Subgenotype 2.1c and 2.1d Strains from China"

_pathogens, 2020, doi:10.3390/pathogens9100821_

Round 1

Reviewer 1 Report

The manuscript by Hao et al. tried to characterize three latest isolates of CSFV in China in vitro and in vivo. This topic is very important for the surveillance and research of CSFV.

I think this manuscript will be stronger if the following points can be explored or clarified:

  1. If the authors could create a table to show the amino acids of three isolates and the control strain in the key position would make the readers easy to follow.
  2. Page 1 Line 36-37, please check “an enveloped RNA virus that is  approximately 12.3 kb in length.
  3. Page 2 Line 51, please check “We report……isolate from…...
  4. Page 2 Line 54, please check “...typical high virulent strain…..
  5. Page 2 Line 61, Here “… tested…..” is OK?
  6. Page 3 Line 108, when we designed this experiment, why did we omit the “PBS/medium only control group” in this study?
  7. Page 3 Line 119, I think “Statistical analysis” is incorrect.
  8. Page 3 Line 130, Please check “…..five generation” .
  9. Page 3 Line 130, Please check “…..five generation”
  10. Page 4 Line 135, “ CSFVs”?
  11. Page 7 Line 170, “yield” is needed?

Author Response

Dear Reviewer:

Thank you for the reviewer’ comments concerning our manuscript entitled “Comparison of the pathogenicity of classical swine fever virus subgenotype 2.1c and 2.1d strains from China” (pathogens-937781). Those comments are all valuable and very helpful for revising and improving our paper. We have studied comments carefully and have made correction which we hope meet with approval. Comments on the presentation of the data is listed below:

  1. If the authors could create a table to show the amino acids of three isolates and the control strain in the key position would make the readers easy to follow.

Response: Thank you very much. Thank you for your careful reading of our manuscript. We have revised the manuscript according to your suggestions.

Table 1. The difference in the amino acid of E2 gene of the isolated virus

Positions (AA)

HCLV

GDGZ-2019

GXGG-2019

HBXY-2019

3

A

S

S

S

20

L

P

P

P

24

G

E

E

E

31

K

K

R

R

34

S

N

S

S

35

Q

H

H

H

36

D

G

G

G

40

N

D

D

D

45

K

R

R

R

49

V

T

T

T

56

T

T

I

I

72

G

R

R

R

75

L

P

P

P

88

N

T

S

S

90

L

V

A

A

91

T

I

I

I

108

S

I

T

T

156

R

K

K

K

158

D

E

E

E

159

K

K

R

R

165

M

V

V

V

166

G

D

D

D

171

T

I

I

I

174

N

R

K

K

182

L

L

W

W

192

E

N

N

N

195

V

T

T

T

197

T

T

M

K

200

V

Q

Q

Q

205

R

R

K

K

212

D

N

N

N

213

G

E

E

E

235

I

V

V

V

249

R

S

S

S

253

S

E

E

E

270

E

G

G

G

273

G

A

G

G

290

M

R

R

R

303

K

R

K

R

305

R

K

K

K

331

A

V

A

A

334

R

H

H

H

336

S

T

T

T

343

V

I

V

V

  1. Page 1 Line 36-37, please check “an enveloped RNA virus that is approximately 12.3 kb in length.”

Response: Thank you very much. We have revised this sentence according to your suggestions (Line 36-37). CSFV is a positive sense single-stranded RNA virus and a member of the genus Pestivirus within the family Flaviviridae.

  1. Page 2 Line 51, please check “We report……isolate from…...”.

Response: Thank you very much. In this study, the characterization of the CSFV isolates from three farms in Hubei, Guangxi, and Guangdong provinces were evaluated (Line 51-52).

  1. Page 2 Line 54, please check “...typical high virulent strain…..”

Response: Thank you very much. To investigate the virulence of subgenotype 2.1c and 2.1d isolates, we compared the pathogenicity of subgenotype 2.1 strains and subgenotype 1.1 SM strain that is the standard high virulent strain in China (Line 54).

  1. Page 2 Line 61, Here “… tested…..” is OK?

Response: Thank you very much. All samples were amplified by reverse transcription-polymerase chain reaction (RT-PCR) by using previously described specific primers, which were used for the CSFV E2 full-sequence gene (Chen et al., 2008) (Line 61-63).

  1. Page 3 Line 108, when we designed this experiment, why did we omit the “PBS/medium only control group” in this study?

Response: Thank you very much. Thank you for your careful reading of our manuscript. The four groups of pigs were housed in the negative-pressure facility of Wuhan Keqian Biology Co., Ltd (Wuhan, China) and were placed in separate rooms to avoid cross infection. All the experimental materials were strictly checked to avoid cross pollution. All CSFV strains were separately cultivated to avoid contamination in P3 biosafe laboratory of Huazhong Agricultural University. All CSFV strains were sequenced to make sure. Pigs in group A (control group) were inoculated intranasally with 2×106 TCID50 of virulent CSFV SM strain. In this experiment, we have set up the positive control.

  1. Page 3 Line 119, I think “Statistical analysis” is incorrect.

Response: Thank you very much. Thank you for your careful reading of our manuscript. We have revised the manuscript according to your suggestions (2.8. Routine blood tests).

  1. Page 3 Line 130, Please check “…..five generation” .

Response: Thank you very much. We have revised the manuscript according to your suggestions. The inoculated cells were passaged successively for the 5th generation.

  1. Page 4 Line 135, “ CSFVs”?

Response: Thank you very much. Thank you for your careful reading of our manuscript.

We have revised the manuscript according to your suggestions.

  1. Page 7 Line 170, “yield” is needed?

Response: Thank you very much. Thank you for your careful reading of our manuscript.

We have deleted the message of ambiguity according to your suggestions.

Reviewer 2 Report

Manuscript review pathogens-937781: “Comparison of the pathogenicity of classical swine fever virus subgenotype 2.1c and 2.1d strains from China.” Corresponding author: Xiangmin Li.

The authors isolate 3 circulating CSFV viruses and characterize their pathogenicity compared to a virulent strain. The manuscript is fairly straightforward and conclusions are justified for the most part, but some changes are required before being accepted for publication.

Major comments:

  1. I believe the title should be reworded. As it stands, it infers that the paper is comparing 2.1c and 2.1d to each other, when the manuscript is a comparison of circulating strains (2.1c and 2.1d) to the pathogenic SM strain.

  1. Figure 4: This figure is described as a one-step growth curve. For this experiment, it is my understanding from the methods that the cells were infected with virus at MOI 0.1, and then viral titers were determined out to 72 hpi. In order to achieve titers of 10^7, the virus would need to continuous replicate, making this a multi-step growth curve.

  1. Methods: Ten clinical parameters are referenced in the methods but only 8 are identified. Furthermore, clinical scoring is highly subjective. Instead of reporting an arbitrary score I think it would strengthen the paper to report the severity of individual symptoms.

  1. Lines 261-262: I don’t think it is accurate to conclude this confirms 2.1c and 2.1d are the major circulating strains in China and that this sentence should be reworded. Three isolates were obtained from field samples that fell into clade 2.1. In addition, the authors reference 2.1c and 2.1d as the dominant strains on line 48 as previously published information.

Minor comments:

  1. Figure 2: ‘Subgroup’ is misspelled.

  1. Methods: Please identify or reference the monoclonal antibodies to E2 used (clone numbers? Previously published? Available commercially). More information is needed.

Author Response

Dear Reviewer:

Thank you for the reviewer’ comments concerning our manuscript entitled “Comparison of the pathogenicity of classical swine fever virus subgenotype 2.1c and 2.1d strains from China” (pathogens-937781). Those comments are all valuable and very helpful for revising and improving our paper. We have studied comments carefully and have made correction which we hope meet with approval.

Major comments:

  1. I believe the title should be reworded. As it stands, it infers that the paper is comparing 2.1c and 2.1d to each other, when the manuscript is a comparison of circulating strains (2.1c and 2.1d) to the pathogenic SM strain.

Response: Thank you very much. Three CSFV isolates were isolated from different provinces of China in the present study. These CSFV isolates belong to two different subgenotypes (2.1c and 2.1d). We compared the pathogenicity of subgenotype 2.1c and 2.1d strains via the intranasal route in weaned piglets. In this experiment, we have set up the positive control (virulent CSFV SM strain, subgenotype 1.1). To study the pathogenicity of CSFV field strains will help us to understand the CSFV biological characteristics and its epidemiology.

  1. Figure 4: This figure is described as a one-step growth curve. For this experiment, it is my understanding from the methods that the cells were infected with virus at MOI 0.1, and then viral titers were determined out to 72 hpi. In order to achieve titers of 10^7, the virus would need to continuous replicate, making this a multi-step growth curve.

Response: Thank you very much. One-step growth curve include the virus replication cycle: latent period, rise phase and stable period. We confirmed that one-step growth experiments in this study is correct. Previous research have applied the method to compare replication kinetics of the CSFV strains (Luo, et al 2017).

Y Luo, S Ji, Y Liu, et al (2017). Isolation and Characterization of a Moderately Virulent Classical Swine Fever Virus Emerging in China. Transbound Emerg Dis. 64(6):1848-1857.

  1. Methods: Ten clinical parameters are referenced in the methods but only 8 are identified. Furthermore, clinical scoring is highly subjective. Instead of reporting an arbitrary score I think it would strengthen the paper to report the severity of individual symptoms.

Response: Thank you very much. Following challenge, the rectal temperature and clinical signs (liveliness, body tension, body shape, breathing, neurological signs, conjunctivitis, appetite, defecation, and so on) were monitored daily. The clinical scoring (CS) system was used to evaluate the virulence of CSFV in accordance with previously established standards (Mittelholzer et al., 2000). Clinical scores were evaluated based on 10 clinical parameters (Mittelholzer et al., 2000). Each parameter were calculated as follows: normal, 0 points; slightly altered, 1 point; distinct clinical signs, 2 point; severe CSF symptom, 3 point. The maximum total score per pig was 30.

The method is a well-established standard to compare the virulence of the CSFV strains. The reference in the manuscript as follows:

Mittelholzer, C., C. Moser, J. Tratschin, and M.A. Hofmann. (2000). Analysis of classical swine fever virus replication kinetics allows differentiation of highly virulent from avirulent strains. Vet Microbiol, 74(4):293-308.

Table 1 Checklist for the determination of the clinical score in CSF animal experiments

No.

Parameter

Criteria

Score

1

Liveliness

Attentive (curious, stands up immediately)

Slightly reduced (stands up hesitantly, but without help)

Tired, gets up only when forced to, lies down again

Dormant, will not stand up

0

1

2

3

2

Body tension

Relaxed, straight back

Stiffness and bent back while standing up, afterwards normal

Bent back and stiff walking remains

Cramps

0

1

2

3

3

Body shape

Full stomach, `round' body

Empty stomach

Empty stomach, thinned body muscles

Emaciated, backbone and ribs visible, head size too big compared to body size

0

1

2

3

4

Breathing (judge before approaching pig)

Frequency 10±15/min, barely visible chest movement

Frequency>20/min

Frequency>20/min, distinct chest movement

Frequency>30/min, breathing through open mouth

0

1

2

3

5

Walking

Well coordinated movements

Hesitant walking, crossed-over legs are corrected slowly

Distinct ataxia/hind lameness, able to walk

Massive lameness, unable to walk

0

1

2

3

6

Skin (in particular ears, nose, legs and tail)

Evenly light pink, hair coat flat

Reddened skin areas

Purple-discolored and cold skin areas, few petechia

Black-red discoloration of skin, no sensitivity, large hemorrhages in skin

0

1

2

3

7

Eyes/conjunctiva

Light pink

Reddened, clear secretion

Highly inflammated, turbid secretion

Highly inflammated, purulent secretion, accentuated blood vessels

0

1

2

3

8

Appetite

Greedy, hungry

Eats slowly when fed

Does not eat when fed, but tastes food

Does not eat at all, shows no interest for food

0

1

2

3

9

Defecation

Soft feces, normal amount

Reduced amount of feces, dry

Only small amount of dry, fibrin-covered feces, or diarrhea

No feces, mucus in rectum, or watery or bloody diarrhea

0

1

2

3

10

Leftovers in feeding trough

Trough empty, clean

Trough almost empty, almost no leftovers

Food only partially eaten

Trough still full, nothing eaten

0

1

2

3

  1. Lines 261-262: I don’t think it is accurate to conclude this confirms 2.1c and 2.1d are the major circulating strains in China and that this sentence should be reworded. Three isolates were obtained from field samples that fell into clade 2.1. In addition, the authors reference 2.1c and 2.1d as the dominant strains on line 48 as previously published information.

Response: Thank you very much. At present, subgenotype 2.1 strains are the dominant pandemic strains in China. Thank you for your careful reading of our manuscript. We have deleted the message of ambiguity according to your suggestions.

Minor comments:

  1. Figure 2: ‘Subgroup’ is misspelled.

Response: Thank you very much. We have revised Figure 2 according to your suggestions.

  1. Methods: Please identify or reference the monoclonal antibodies to E2 used (clone numbers? Previously published? Available commercially). More information is needed.

Response: Thank you very much. CSFV E2 proteins were expressed in baculovirus expression systems (Zhang H, Li X, Peng G, Tang C, Zhu S, Qian S, et al. Glycoprotein E2 of classical swine fever virus expressed by baculovirus induces the protective immune responses in rabbits. Vaccine 2014;32.). AH09 E2-specific monoclonal antibodies were produced in our lab.

Reviewer 3 Report

The manuscript is well written. The protocols used are adequately described.

Conclusions could be a little more focused on the importance of characteize  the CSF strains in order to improve the control plans.

Author Response

Dear Reviewer:

Thank you for the reviewer’ comments concerning our manuscript entitled “Comparison of the pathogenicity of classical swine fever virus subgenotype 2.1c and 2.1d strains from China” (pathogens-937781). The comment is valuable and very helpful for revising and improving our paper. We have studied comment carefully and have made correction which we hope meet with approval.

The comment:

Conclusions could be a little more focused on the importance of characteize the CSF strains in order to improve the control plans.

Response: Thank you very much. We have revised the conclusion according to your suggestions. At present, subgenotype 2.1 strains are the dominant pandemic strains in China [6, 10, 23]. Three CSFV isolates were isolated from different provinces of China in the present study. These CSFV isolates belong to two different subgenotypes (2.1c and 2.1d). We compared the pathogenicity of subgenotype 2.1c and 2.1d strains via the intranasal route in weaned piglets. After pigs infected CSFV 2.1 isolates, there were marked some characteristic changes, such as hemocytopenia, especially leukopenia and thrombocytopenia. CSFV SM-infected pigs showed reduced leukocyte and PLT counts until death. However, leukopenia and thrombocytopenia of GDGZ-2019, HBXY-2019, and GXGG-2019 infected pigs were significantly less than those in CSFV SM-infected pigs and gradually recovered and went back to normal at the end of the experiment. CSFV 2.1c (GDGZ-2019) and 2.1d (HBXY-2019 and GXGG-2019) strains are moderate virulent strains, similar to the published results on subgenotype 2.1 strains. This study shows the necessity of monitoring the molecular epidemiology and the etiological characteristics of the epidemic CSFV 2.1 isolates, which may help us to understand the CSFV 2.1 isolates biological characteristics and control the CSF outbreaks.